# LLM as a Classifier: Leveraging Large Language Models for Text and Vision Classification

## Abstract

Classification is a fundamental capability for AI systems, yet current large language model (LLM) approaches remain poorly suited for latency-critical applications. Prompting and constrained decoding produce verbose, multi-token outputs that require expensive token-by-token generation, while encoder-based models achieve faster inference at the cost of flexibility and generative capacity. We propose LaaC (LLM as a Classifier), a framework that formulates classification as constrained generation with single-token outputs. By introducing atomic label tokens and applying parameter-efficient fine-tuning, our method reduces classification to a deterministic one-step decoding problem. Experiments across text and multimodal benchmarks demonstrate both strong accuracy and consistently fast inference. On MIntRec 2.0, a fine-tuned Gemma-3-27B model attains 62.7% accuracy, outperforming GPT-4o (43.7%) and GPT-5 (51.8%) while running more than an order of magnitude faster. On standard text classification benchmarks, our models match GPT-4o in accuracy while achieving $8\times$ lower tail latency. These results establish decoder-style LLMs as practical and scalable classifiers for real-time applications. Our code is available at `https://anonymous.4open.science/r/LaaC_ICLR`.

## 1 Introduction

Classification is a fundamental task in machine learning (Sebastiani, 2002) with widespread applications across domains, from sentiment analysis (Pang et al., 2008) and intent recognition (Goo et al., 2018; Chen et al., 2019) to customer support and interactive dialogue agents. As these applications increasingly operate on multimodal data—combining text and vision—there is growing demand for unified models that can handle diverse input modalities while maintaining efficiency in latency-sensitive environments (Wang et al., 2024).

Current approaches to classification with large language models (LLMs) face significant limitations, particularly in latency-critical applications. Prompt-based methods, while intuitive, often produce verbose, multi-token responses that require additional parsing and introduce substantial inference overhead. More importantly, they provide no guarantee that outputs will be single tokens: a request such as "classify this review as positive or negative" can yield explanatory sentences or multi-token paraphrases rather than clean categorical labels. Even with constrained decoding techniques (Geng et al., 2023) that restrict outputs to valid label strings, models still rely on token-by-token generation. This scales poorly with label vocabulary size and leads to unpredictable latency variations.

This latency challenge is particularly acute in real-world deployment scenarios where classification must occur at scale with strict response time requirements. Traditional encoder-based approaches (e.g., BERT with classification heads) offer predictable, low-latency inference but lack the flexibility and generative capabilities that make modern LLMs attractive for complex reasoning tasks, since they require task-specific architectures and dataset-specific fine-tuning.

In this work, we propose LaaC (LLM as a Classifier), an approach that bridges this gap by treating classification as a constrained generation task with **single-token outputs**. Our key insight is to introduce atomic special tokens (e.g., `[control_1]`) for each class, enabling the model to produce decisions in exactly one generation step. As illustrated in Figure 1, this design not only eliminates

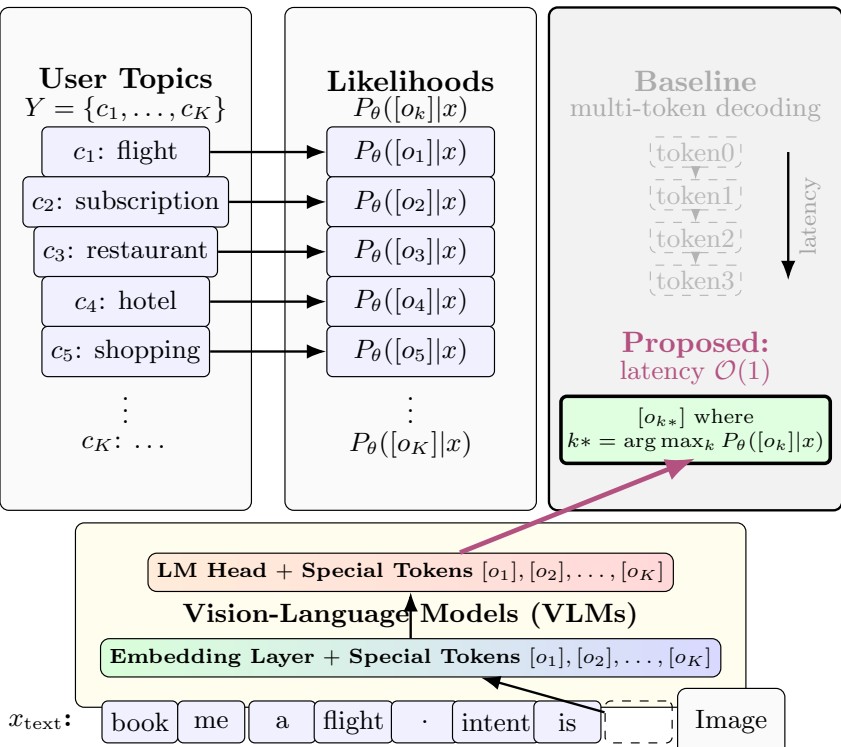

Figure 1: **Overview of our LaaC framework.** Inputs are processed by a decoder-style LLM that directly outputs an atomic special token for the target class. Unlike baseline prompting with multi-token decoding, the framework guarantees $\mathcal{O}(1)$ latency and supports zero-shot adaptation.

multi-token decoding overhead but also enables *zero-shot classification*: by reassigning label tokens at inference time, the model seamlessly adapts to new tasks without task-specific retraining.

We demonstrate this approach using parameter-efficient fine-tuning (LoRA) on vision-language models, creating a unified framework that handles both text-only and multimodal classification tasks. Our method rests on three central pillars: **accuracy**, **latency**, and **generality**. By constraining the output space to a finite set of learned tokens, we achieve deterministic single-step inference while maintaining the semantic understanding capabilities of large pretrained models. Our contributions are threefold:

1. **A unified single-token classification framework** that treats classification as constrained generation, eliminating multi-token decoding inefficiencies while preserving the generalization of decoder-based LLMs.

2. **Significant latency improvements** through atomic label tokens that enable deterministic single-step inference, achieving predictable sub-second inference and maintains efficiency as label spaces grow.

3. **Strong empirical results** across diverse benchmarks, our fine-tuned Gemma-3-27B model outperforms GPT-4o, GPT-5, and encoder-based baselines on multimodal evaluation. On text classification benchmarks, our models remain competitive.

Our approach proves that with careful design, decoder-based models can achieve both the effectiveness and efficiency of specialized encoder architectures while maintaining their broader generative capabilities, making them practical for latency-sensitive applications.

## 2 RELATED WORK

### 2.1 PROMPT-BASED AND FEW-SHOT CLASSIFICATION WITH LLMS

One natural way to adapt large language models (LLMs) for classification is through *prompting*. For example, a model can be asked: ``Classify the following review as positive or negative: {text}''. While simple, this approach often yields verbose, multi-token answers rather than a clean class symbol. *Few-shot prompting* (Brown et al., 2020) improves reliability by adding in-context demonstrations, and CARP (Sun et al., 2023) further refines this by encouraging models to extract clues and reason step by step. Instruction tuning further enhances this paradigm by aligning models to follow task instructions more robustly (Ouyang et al., 2022; Wei et al., 2021), and multimodal prompting has been demonstrated in vision–language models such as CLIP (Radford et al., 2021). Despite these advances, the outputs of prompting-based methods remain free-form text sequences, which complicates integration into structured applications and hinders efficiency.

To mitigate this, methods such as PET and LM-BFF reformulate classification as a cloze task with *verbalizers* that map each class to a natural-language string and fine-tune the model on small labeled sets (Schick & Schütze, 2020; Gao et al., 2020). Beyond discrete verbalizers, *continuous prompting* methods have been proposed: Prompt Tuning learns soft prompt embeddings optimized for a target task (Ding et al., 2023), while Prefix Tuning prepends trainable key–value vectors to each transformer layer (Li & Liang, 2021). Although effective, these approaches are typically developed for few-shot learning scenarios with limited labeled data; in contrast, our work targets settings with more training examples, where parameter-efficient fine-tuning can be applied to adapt large models.

Recent efforts have also explored *constrained decoding*, where logits are masked or grammar rules are applied so that only valid label strings can be produced (Geng et al., 2023). While this ensures format consistency, it introduces longer latency (due to token-by-token decoding even for short label strings) and limits flexibility when scaling to large label spaces or multimodal tasks. These drawbacks motivate our design of a *finite set of atomic label tokens*, which reduces classification to a single constrained generation step and eliminates the inefficiencies of multi-token verbalizers.

### 2.2 ENCODER-BASED FINE-TUNING FOR CLASSIFICATION

A widely adopted paradigm emphasizes *fine-tuning encoder models* with a dedicated classifier head. Transformer encoders such as BERT and its successors (e.g., RoBERTa, DeBERTa) have become the standard for text classification: the input sequence is processed by the encoder, and the contextualized representation of the [CLS] token is passed through a linear layer trained with cross-entropy loss (Devlin et al., 2019; Liu et al., 2019; He et al., 2020). For multimodal classification, encoder-style fusion models extend this paradigm by incorporating vision or audio encoders and cross-modal attention modules, as in MulT and MAG-BERT (Tsai et al., 2019; Rahman et al., 2020). These approaches are efficient and effective, but they require task-specific classifier heads and lack the flexibility of decoder-based LLMs. In contrast, our method shows that *decoder-style VLMs*, adapted with parameter-efficient fine-tuning (LoRA) and single-token label spaces, can match or surpass encoder baselines on challenging multimodal datasets while maintaining compatibility with generative tasks and supporting a broad range of downstream applications.

## 3 METHODOLOGY

### 3.1 PROBLEM FORMULATION

We formalize classification as a supervised learning problem. Each training instance consists of an input tuple

$$x = \left(x^{\text{text}},\, x^{\text{vis}}\right),$$

where $x^{\text{text}}$ denotes the textual input and $x^{\text{vis}}$ represents the vision modality, which may include static images or short video segments. Not all instances contain both modalities; missing modalities are treated as empty. The objective is to learn a mapping

$$f_\theta : \left(x^{\text{text}},\, x^{\text{vis}}\right) \,\mapsto\, y,$$

where $y \in \mathcal{Y}$ is a categorical label drawn from a predefined set of classes $\mathcal{Y} = \{c_1, c_2, \ldots, c_K\}$.

Given a dataset

$$\mathcal{D} = \{(x_i, y_i)\}_{i=1}^{N},$$

the learning goal is to estimate model parameters $\theta$ that minimize the expected classification loss:

$$\theta^* = \arg\min_{\theta} \ \mathbb{E}_{(x,y)\sim\mathcal{D}} \ \mathcal{L}\big(f_\theta(x), y\big),$$

where $\mathcal{L}$ is the cross-entropy loss over labels. This formulation generalizes unimodal classification to a multimodal setting, where textual and visual inputs are jointly leveraged for improved prediction.

## 3.2 Model architecture and fine-tuning strategy

Our proposed framework LaaC is model-agnostic and can be applied to a wide range of large language models and vision-language models, including Gemma, Mistral and Qwen architectures. The general principle is to treat classification as a constrained generation task, where the model is guided to produce a single special token corresponding to the target label.

**Parameter-Efficient Fine-Tuning.** We adopt Low-Rank Adaptation (LoRA) as the primary fine-tuning strategy (Hu et al., 2022). LoRA introduces a small number of trainable rank-decomposition matrices into the attention and projection layers, while keeping the majority of model parameters frozen. This approach enables efficient adaptation across different base models with significantly reduced memory and compute requirements.

**Special Tokens for Labels.** For each classification category $c_k \in \mathcal{Y}$, we introduce a unique special token $[o_k]$. Conventional choices such as digits or short text labels are problematic: they can appear naturally in the input, causing ambiguity, and larger indices (e.g., "100") are split into multiple subword tokens, leading to multi-step decoding and higher latency. To provide grounding, a natural-language description of each class is included in the system prompt (see Appendix A.2 for prompt templates). During fine-tuning, the model is trained to map $\big(x^{\text{text}}, x^{\text{vis}}\big)$ directly to the correct special token. In implementation, we extend the tokenizer with 500 reserved control tokens $[o_1], \ldots, [o_{500}]$, which allows the model to accommodate up to 500 classes during inference without retraining.

**Training Objective.** The training objective is defined as cross-entropy loss over the predicted control token. Since classification requires only a single output, the loss is computed exclusively on the final assistant response token (e.g., $[o_k]$), while all preceding tokens in the input sequence are masked out and excluded from loss calculation. Formally, for each instance $(x, k)$, the loss is

$$\mathcal{L} = -\log P_\theta\big([o_k] \,\big|\, x\big).$$

## 3.3 Special Token Design (Single-Token Outputs)

Let $\mathcal{Y} = \{c_1, \ldots, c_K\}$ denote the label set and let $\mathcal{V}$ be the base tokenizer vocabulary. We augment the vocabulary with a set of $K$ *atomic output symbols*:

$$\Omega = \{[o_1], [o_2], \ldots, [o_K]\}, \qquad \tilde{\mathcal{V}} = \mathcal{V} \cup \Omega.$$

Each $[o_k]$ is a dedicated *control token* corresponding to class $c_k$, with a trainable embedding $e_k \in \mathbb{R}^d$. These rows are appended to the model's embedding matrix and are updated during fine-tuning, together with LoRA-adapted weights.

**Why single-token outputs.** If labels are represented as natural-language strings $s_k$ (e.g., `"book a flight"`), a subword tokenizer typically produces a variable-length sequence

$$\tau(s_k) = (v_1, \ldots, v_{m_k}), \quad m_k \geq 1,$$

where $\tau(\cdot)$ denotes the tokenization function. This variability requires loss over multiple decoding steps and depends on segmentation. In contrast, our design assigns each class a single atomic symbol $\omega_k = [o_k] \in \Omega$, which is guaranteed to decode as exactly one token. This yields three benefits: (i) the output space collapses to $K$ symbols, (ii) ambiguity from label strings that may appear in the input is eliminated, and (iii) decoding and evaluation are simplified to a deterministic one-step classification.

**Randomized label assignments.** To prevent memorization of static token–label associations, we do not fix a permanent mapping between special tokens and semantic classes. Instead, during preprocessing we randomly shuffle the correspondence between classes and control tokens across training instances. This design encourages the model to rely on the contextual descriptions of labels provided in the prompt, rather than memorizing token identities. As a result, the model learns to infer the correct output token from the input context, which improves robustness and generalization across datasets and label spaces. Without randomization, models tended to overfit to token IDs and failed to generalize to new mappings.

**Separation from context.** We reserve a sentinel namespace for $\Omega$, ensuring that these tokens are never decomposed into subwords and never occur in the input text. Formally, $\Omega \cap \mathcal{V} = \emptyset$ and $\Omega \cap \tau(x) = \emptyset$ for any input $x$. In practice, this is enforced by registering $[o_k]$ as special tokens in the tokenizer, so that they are available to the model during output generation.

**Training objective (single position).** Given input $x = \left(x^{\text{text}}, x^{\text{vis}}\right)$ with gold label $k \in \{1, \ldots, K\}$, the model is required to emit exactly one control token $[o_k]$. Let $h_T$ denote the decoder state at the output position (the single assistant step). We compute logits restricted to $\Omega$:

$$z_j = (Wh_T + b)_j, \qquad P_\theta([o_j] \mid x) = \frac{\exp(z_j)}{\sum_{i=1}^{K} \exp(z_i)} \quad \text{for } j \in \{1, \ldots, K\}.$$

The loss is standard cross-entropy over this single prediction:

$$\mathcal{L}(x, k) = -\log P_\theta([o_k] \mid x).$$

All preceding tokens (system/user prompts and any in-context descriptions) are masked out of the loss, so classification supervision is concentrated solely on the final output position. In implementation, we construct a binary loss mask: the position of the response token is assigned its class label, while all other positions are set to $-100$, which the cross-entropy loss ignores.

**Inference rule.** At test time, decoding reduces to a single restricted argmax:

$$k^* = \arg \max_{k \in \{1, \ldots, K\}} P_\theta([o_k] \mid x).$$

Equivalently, we set `max_new_tokens = 1` and restrict the decision space to $\Omega$, ensuring the model outputs exactly one control token (e.g., $[o_{k^*}]$) rather than a multi-token string. This deterministic procedure avoids token-by-token generation and guarantees constant-time inference.

## 4 EXPERIMENTS

### 4.1 DATASETS

**Training corpus.** We construct a balanced training corpus of 28k examples, comprising 14k text-only and 14k multimodal classification instances. **Text datasets.** The text-only portion includes examples, each consisting of a natural language input paired with a categorical label. The corpus spans diverse domains, specifically, we use the CLINC dataset (Larson et al., 2019) (150 intents across 10 domains, with out-of-scope examples), the AMZN-MASSIVE dataset (FitzGerald et al., 2022; Bastianelli et al., 2020) (60 intents spanning 18 domains), and MULTIWOZ-2.2 (Zang et al., 2020) (11 intents across 3 domains). To encourage generalization beyond standard datasets, we additionally incorporate constrained instruction-following data from FollowBench and Unnatural Instructions (Jiang et al., 2023; Honovich et al., 2022), as well as synthetic data generated with Agent-Gym (Xi et al., 2024). **Multimodal datasets.** The multimodal portion comprises 14k examples from established vision–language datasets. We include video–text pairs from MINTREC (Zhang et al., 2022) and reformulate image–question pairs from subsets of A-OKVQA (Schwenk et al., 2022) and VISUAL7W (Zhu et al., 2016). To construct this corpus, we sample approximately 5k examples each from the image-based datasets while retaining all available MINTREC frames. All data are reformulated into a unified JSON schema (see Fig. 2), with consistent `messages` fields and an optional `image_path`. To mitigate label memorization, we randomize control-token assignments: class labels are mapped to tokens drawn from a reserved set of 500 control tokens.

```
Example of a Classification Training Instance

 {
"messages": [
  { "role":  "system", "content":  "You are a classification
expert.  Topics:  [control_x] Complain, [control_y] Praise,
[control_z] Apologise ..." },
  { "role":  "user", "content":  "### USER CONVERSATION HERE ###" },
  { "role":  "assistant", "content":  "[control_x]" }
],
"image_path":  "path/to/image.jpg",
}
```

Figure 2: Illustration of a fine-tuning training instance in our classification datasets. Each sample includes the structured `messages` field and optional `image_path`.

**Evaluation benchmarks.** We evaluate our approach on a diverse suite of multimodal and text-only classification benchmarks. For multimodal evaluation, we use the official test split of **MIntRec2.0** (Zhang et al., 2024), a large-scale benchmark for intent recognition in multimodal dialogues that combine text and vision. This dataset includes 30 fine-grained intent classes and requires reasoning over conversational context and multiple modalities, making it a challenging test of real-world multimodal classification. For text-only tasks, we consider four widely used benchmarks: **SST-2** (Socher et al., 2013) is a binary sentiment analysis benchmark consisting of movie reviews annotated as positive or negative; **Amazon Reviews Polarity** (McAuley & Leskovec, 2013; Zhang et al., 2015) contains millions of product reviews labeled as positive or negative. It evaluates sentiment classification in a large-scale, noisy e-commerce domain; **AG News** (Zhang et al., 2015) is a topic classification dataset with four categories: World, Sports, Business, and Science/Technology; and **DBpedia** (Lehmann et al., 2015) is a 14-class benchmark built from Wikipedia articles, covering categories such as Company, Artist, Athlete, and Place. It provides a broad test of factual and encyclopedic text classification. Example prompt templates for these datasets are provided in Appendix A.2 (Figures 4 to 8). For text-only benchmarks, we evaluate on 200 randomly sampled test examples from each dataset to ensure consistent and efficient comparisons across models.

## 4.2 BASELINES

We benchmark against both open-source and proprietary systems.

**Pretrained LLMs.** We include the untuned versions of Gemma-3 (4B, 27B) and Mistral-3-24B as base VLM checkpoints. These serve as reference points for the capability of large pretrained models without classification adaptation.

**External API models.** For stronger upper-bound comparisons, we evaluate proprietary multimodal models including GPT-4O and GPT-5 (including GPT-5-NANO). These systems represent state-of-the-art commercial offerings.

**Encoder-based models.** To contextualize against specialized architectures, we also report results for strong encoder-style multimodal baselines (e.g., MAG-BERT and MulT), following the MINTREC 2.0 benchmark protocol. For text benchmarks, we include: (i) a traditional linear classification head applied to BERT-base and RoBERTa-base (16-shot fine-tuning), and (ii) LM-BFF (Gao et al., 2020) with a RoBERTa-base backbone as a representative prompt-based few-shot classifier.

## 4.3 TRAINING DETAILS

We fine-tuned both Gemma-3 and Mistral-3 with a batch size of 1 and gradient accumulation of 16 (effective batch size 16). Models were trained for 30 epochs using a learning rate of $2 \times 10^{-5}$ with a warmup ratio of 0.1. LoRA modules (rank 8, $\alpha = 16$, dropout 0.05) were applied to attention and feed-forward layers. Gemma-3 employed tied embeddings, whereas Mistral-3 used untied embeddings that required explicit saving. Training was conducted on $8 \times$NVIDIA A100 GPUs with mixed precision (bfloat16), gradient checkpointing, and DeepSpeed ZeRO-3. Early stopping was applied

with a patience of 8 validation steps and a minimum improvement threshold of $10^{-4}$. Model performance was evaluated every 500 steps using validation loss as the criterion. Further implementation details are provided in Appendix A.3.

## 4.4 EVALUATION METRICS

We evaluate models along two complementary dimensions: classification effectiveness and inference efficiency. **Classification accuracy.** The primary metric is accuracy, computed as the percentage of instances where the predicted control token matches the gold label. Accuracy is reported separately for each dataset. **Latency.** To capture efficiency, we measure model response times. We report the median latency (P50) and the tail latency (P95) across evaluation batches. All baselines are evaluated with our vLLM-based inference framework (Kwon et al., 2023) on a single NVIDIA A100 GPU, using consistent input formatting and datasets. All latency measurements reported in the main paper use batch size 1 to ensure a controlled and consistent comparison across models. We further examine how latency scales with batch size, and Appendix A.7 presents a batch-scaling study demonstrating that the latency benefits of LaaC persist even under large-batch inference settings.

## 4.5 RESULTS

### 4.5.1 MULTIMODAL EVALUATION RESULTS

We evaluate our fine-tuned models trained on the multimodal portion of our corpus on the challenging MIntRec 2.0 dataset, which requires understanding multimodal dialogue contexts (text, image, video) for intent recognition. The evaluation covers both base models and fine-tuned variants of Gemma-3 and Mistral-3, with comparisons against GPT models such as GPT-4o and GPT-5, as well as encoder-based baselines reported in the original paper.

**MIntRec 2.0 (Multimodal Topic Classification).** Table 1 presents results on MIntRec 2.0. Base Gemma-3 models perform poorly ($\sim$16–18% accuracy), underscoring the difficulty of multimodal intent recognition without adaptation. After fine-tuning, performance improves markedly: Gemma-3-4B reaches **55.2%**, while Gemma-3-27B achieves **62.7%**, significantly outperforming GPT-4o (43.7%) and GPT-5 (51.8%) while remaining much faster. Fine-tuned Gemma-3 models also achieve low latency (P95 < 1s), in contrast to GPT-4o and GPT-5, which require 6–13s.

Beyond this standard GPT-4o configuration, we also explore whether carefully designed class-index prompts can emulate our single-token interface; Appendix A.4 (*Alternative Prompting Baselines*) shows that such prompting-only variants still frequently produce multi-token outputs and incur additional decoding latency. To further strengthen this comparison, we additionally evaluate a constrained-decoding variant of GPT-4o, where all 30 intents are mapped to digit-only labels and decoding is restricted to this label set. This improves GPT-4o's accuracy to 45.56% and reduces its median latency from 4.30 s to 2.98 s, but it still remains an order of magnitude slower than LaaC-adapted open-source models, which operate with deterministic single-token outputs (P50 $\leq$ 0.37 s).

**Comparison with Encoder-Based Models.** MAG-BERT and MulT, strong encoder-based multimodal baselines from the original MIntRec paper, achieve around 60.6% accuracy. Our fine-tuned Gemma-3-27B surpasses both, reaching **62.7%**, while also offering the benefits of a unified generative modeling framework. When evaluated end-to-end in our pipeline, MAG-BERT and MulT exhibit competitive latency (e.g., P50 = 0.23–0.24 s, P95 = 0.40–0.41 s), comparable to Gemma-3-4B (FT, LaaC). However, these encoder-based models are fully fine-tuned specifically on MIntRec 2.0 with task-specific classifier heads tied to a fixed label space, and therefore do not generalize to new tasks or label sets without retraining. In contrast, LaaC models preserve the generative interface of decoder LLMs and support zero-shot adaptation across tasks; we further analyze this generality in Sec. 4.5.2. The competitive or superior performance of fine-tuned LLMs indicates that large decoder-based architectures, when fine-tuned on in-domain data, can match or outperform specialized encoder-based models while simultaneously enabling broader generative and reasoning capabilities. To better contextualize these baselines, we additionally examine the end-to-end latency of the MAG-BERT and MulT pipelines; the detailed breakdown is provided in Appendix A.5.

Table 1: MIntRec 2.0 evaluation results, including baselines from the original paper ($^\diamond$) and our evaluations. FT = fine-tuned. Models are sorted by accuracy. Speedup is relative to GPT-4o (P50 = 4.30s, P95 = 6.12s).

| Model | Accuracy (%) | P50 (s) | Speedup (P50) | P95 (s) | Speedup (P95) |
|---|---|---|---|---|---|
| Gemma-3-4B (Base) | 16.04 | 1.33 | 3.23× | 1.77 | 3.46× |
| Gemma-3-27B (Base) | 17.76 | 2.18 | 1.97× | 2.77 | 2.21× |
| Mistral-3-24B (Base) | 40.83 | 0.77 | 5.58× | 1.71 | 3.58× |
| GPT-5-nano | 41.02 | 3.67 | 1.17× | 5.46 | 1.12× |
| GPT-4o | 43.68 | 4.30 | 1.00× | 6.12 | 1.00× |
| GPT-4o (w/ Constrained Decoding) | 45.56 | 2.98 | 1.44× | 5.85 | 1.05× |
| **Mistral-3-24B (FT, LaaC)** | **49.34** | **0.64** | **6.72×** | **1.64** | **3.73×** |
| GPT-5 | 51.84 | 7.13 | 0.60× | 13.01 | 0.47× |
| **Gemma-3-4B (FT, LaaC)** | **55.19** | **0.26** | **16.54×** | **0.60** | **10.20×** |
| MAG-BERT$^\diamond$ | 60.58 | 0.23 | 18.70× | 0.40 | 15.30× |
| MulT$^\diamond$ | 60.66 | 0.24 | 17.92× | 0.41 | 14.93× |
| **Gemma-3-27B (FT, LaaC)** | **62.72** | **0.37** | **11.62×** | **0.90** | **6.80×** |

### 4.5.2 TEXT CLASSIFICATION ACROSS DOMAINS

Our evaluations span four widely used classification benchmarks: SST-2, Amazon Reviews, AG News, and DBpedia. Evaluations on additional benchmarks (TweetTopic and Banking77) are provided in Appendix B. Unlike encoder-based baselines (e.g., BERT or RoBERTa variants) and prompt-based few-shot methods such as LM-BFF, which are typically fine-tuned directly on each dataset and thus achieve strong but task-specific results, our framework is *not fine-tuned on any of these datasets*. Instead, we evaluate zero-shot generalization by comparing our fine-tuned classifier against its untuned base model, GPT-4o, and encoder-style baselines. The results are in Table 2.

As a basic encoder-style baseline, we include BERT-base and RoBERTa-base models equipped with a linear classification head, fine-tuned using 16 examples per class. As shown in Table 2, these linear-head models achieve very low latency but consistently exhibit the weakest accuracy across all four benchmarks, trailing LaaC by a substantial margin. To provide a stronger encoder-based comparison, LM-BFF augments RoBERTa with prompt-based verbalizers and in-context examples. LM-BFF clearly improves over linear-head fine-tuning across all datasets, reaching 87.5–90.0% accuracy on Amazon Reviews, AG News, and DBpedia. However, it still falls short of our LaaC models, which achieve up to 95.5% on SST-2, 94.0–95.0% on Amazon Reviews and DBpedia. Moreover, LM-BFF incurs higher latency than linear-head models (approximately 0.14–0.16 s), and like the linear-head baselines, it requires specific fine-tuning for each dataset and label space.

Without task-specific adaptation, LaaC demonstrates **robust cross-domain performance**. As shown in Table 2, on sentiment classification (SST-2, Amazon Reviews) and encyclopedic categorization (DBpedia), our fine-tuned models achieve accuracy comparable to GPT-4o and close to its base model, while maintaining **consistently sub-200 ms inference latency** (P95 $\leq$ 0.13 s). Most notably, the efficiency gains are substantial. While GPT-4o requires close to one second for tail latency on these tasks, our approach reduces this by nearly an order of magnitude. By collapsing classification into a deterministic single-token decision, inference time becomes both **fast and predictable**, which is crucial for latency-sensitive deployments.

Overall, these results highlight that our approach **generalizes well across unseen text domains without any task-specific fine-tuning**. In contrast to encoder models that trade flexibility for efficiency, our method preserves the generative capacity of large decoder LLMs while matching their classification accuracy and surpassing them in efficiency. Beyond cross-dataset generalization, we also examine whether the model can operate under unseen mappings between semantic labels and special control tokens; the corresponding analysis and results are provided in Appendix A.6. A consolidated three-axis comparison of accuracy, latency, and generality is provided in Appendix C, offering a Pareto-style view of model performance across these dimensions.

Table 2: Text-only evaluation results on SST-2, Amazon Reviews, AG News, and DBpedia. We report accuracy and latency (median *P50* and tail *P95*).

| Dataset | Model | Acc. (%) | P50 Latency (s) | P95 Latency (s) |
|---|---|---|---|---|
| SST-2 | BERT Linear Head | 70.00 | 0.007 | 0.008 |
| | RoBERTa Linear Head | 64.50 | 0.008 | 0.008 |
| | LM-BFF | 89.33 | 0.15 | 0.16 |
| | GPT-4o | 95.50 | 0.53 | 0.84 |
| | Mistral-3-24B (Base) | 95.50 | 0.07 | 0.07 |
| | **Mistral-3-24B (FT, LaaC)** | 95.50 | **0.03** | **0.03** |
| | Gemma-3-27B (Base) | 95.00 | 0.36 | 0.41 |
| | **Gemma-3-27B (FT, LaaC)** | 95.50 | **0.11** | **0.12** |
| Amazon Reviews | BERT Linear Head | 77.00 | 0.007 | 0.008 |
| | RoBERTa Linear Head | 78.50 | 0.007 | 0.008 |
| | LM-BFF | 87.50 | 0.14 | 0.15 |
| | GPT-4o | 95.00 | 0.53 | 0.97 |
| | Mistral-3-24B (Base) | 95.50 | 0.08 | 0.09 |
| | **Mistral-3-24B (FT, LaaC)** | 95.00 | **0.05** | **0.08** |
| | Gemma-3-27B (Base) | 93.50 | 0.38 | 0.46 |
| | **Gemma-3-27B (FT, LaaC)** | 94.00 | **0.10** | **0.12** |
| AG News | BERT Linear Head | 79.50 | 0.007 | 0.008 |
| | RoBERTa Linear Head | 79.50 | 0.007 | 0.008 |
| | LM-BFF | 89.00 | 0.14 | 0.15 |
| | GPT-4o | 84.50 | 0.55 | 1.00 |
| | Mistral-3-24B (Base) | 84.00 | 0.07 | 0.17 |
| | **Mistral-3-24B (FT, LaaC)** | 83.00 | **0.05** | **0.05** |
| | Gemma-3-27B (Base) | 84.00 | 0.25 | 0.60 |
| | **Gemma-3-27B (FT, LaaC)** | 81.50 | **0.11** | **0.13** |
| DBpedia | BERT Linear Head | 91.50 | 0.007 | 0.008 |
| | RoBERTa Linear Head | 82.50 | 0.008 | 0.008 |
| | LM-BFF | 90.00 | 0.15 | 0.15 |
| | GPT-4o | 97.00 | 0.61 | 1.23 |
| | Mistral-3-24B (Base) | 94.50 | 0.10 | 0.21 |
| | **Mistral-3-24B (FT, LaaC)** | 93.00 | **0.05** | **0.06** |
| | Gemma-3-27B (Base) | 97.00 | 0.25 | 0.48 |
| | **Gemma-3-27B (FT, LaaC)** | 95.00 | **0.10** | **0.11** |

## 4.6 EFFECT OF LABEL-SET SIZES

We further analyze the impact of the number of label sets on model performance by evaluating across datasets with increasing topic sizes, ranging from binary sentiment classification (SST-2, Amazon Reviews) to multi-class categorization (AG News with 4 topics and DBpedia with 14 topics).

**Generalization ability.** As shown in Figure 3a, our fine-tuned Gemma-3-27B model consistently maintains high accuracy across datasets of varying difficulty. Even as the label space expands from 2 to 14 categories, the accuracy remains 95% on DBpedia and comparable to binary sentiment datasets, demonstrating strong zero-shot generalization ability.

**Efficiency stability.** In addition to accuracy, we examine efficiency via P50 latency. The results reveal that latency remains remarkably stable across datasets, fluctuating only within 0.10–0.11s despite the growth in label space. This indicates that our design achieves scalable inference efficiency while handling tasks of increasing complexity.

## 4.7 SCALING ANALYSIS

We further investigate how accuracy and efficiency scale with model sizes. Figures 3b and 3c report results for Gemma-3 models with 4B, 12B, and 27B parameters on four text benchmarks. **Accuracy.** Performance improves consistently as model size increases (Figure 3b). The 4B model shows lower accuracy, particularly on AG News, while the 12B model closes much of the gap. The 27B model achieves the strongest results across all datasets, exceeding 95% on SST-2 and DBpedia and remain-

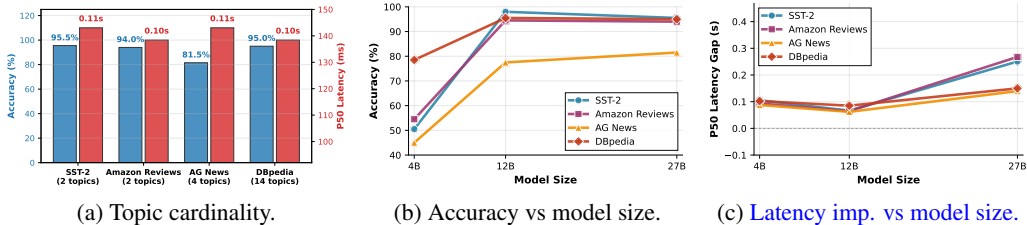

(a) Topic cardinality.      (b) Accuracy vs model size.      (c) Latency imp. vs model size.

Figure 3: Performance of Gemma-3 models across datasets.

ing above 94% on Amazon Reviews. **Latency.** Figure 3c reports the *P50 latency improvement* of LaaC relative to the corresponding base model. The improvement is positive across all datasets, showing that LaaC consistently reduces latency compared to standard autoregressive decoding. The improvement is modest for the 4B and 12B models, but becomes clearly larger for the 27B model. This trend indicates that LaaC's efficiency advantage becomes more pronounced for larger models.

### 4.8 EFFECT OF TEXT TRAINING DATA

To assess the value of incorporating both unimodal and multimodal supervision, we evaluated fine-tuned Gemma-3-27B on a held-out proprietary dataset. When trained only on multimodal data, the model achieved 80.6% accuracy with P50 = 0.06s and P95 = 0.18s. By contrast, when training combined both text-only and multimodal data, accuracy improved to 84.7% while maintaining comparable efficiency (P50 = 0.06s, P95 = 0.06s). These results suggest that exposing the model to both text and multimodal supervision during training provides stronger representations and leads to consistent accuracy gains. A similar improvement ($\approx$0.5%) at unchanged latency was also observed on DBpedia, indicating that the effect generalizes beyond a single dataset.

## 5 CONCLUSION

We introduced a framework LaaC that treats classification as a constrained generation problem with *single-token* outputs. By augmenting decoder-style LLMs with atomic label tokens and adapting them through parameter-efficient fine-tuning, our method collapses classification into a deterministic one-step decoding task. This design achieves significant latency reductions while preserving the generative flexibility of large models. Empirically, fine-tuned Gemma-3 models outperform much larger proprietary systems on the MIntRec 2.0 benchmark and match or surpass encoder-based multimodal baselines, all while running with sub-second tail latency. These results demonstrate that decoder LLMs can serve as practical, scalable classifiers for latency-sensitive applications.

Our current study focuses on text and vision inputs; extending the framework to additional modalities such as audio remains an open direction. While our evaluation highlights substantial latency improvements, a deeper analysis of calibration, robustness, and multilingual generalization is needed to validate deployment readiness. Future work will explore scaling to larger label spaces, integrating rejection mechanisms for out-of-scope detection, and combining single-token classification with reasoning-augmented LLMs. Together, these directions aim to advance LLMs as both versatile generators and efficient classifiers for real-world multimodal systems.

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

# A APPENDIX

## A.1 LLM USAGE DISCLOSURE

We used large language models (e.g., ChatGPT) only to polish writing and assist with literature search. They were not used for generating research ideas or results.

```
Example Prompt (SST-2)

 {
"messages":  [
  { "role":  "system", "content":  "You are a classification
expert.  Topics:  [control_1] Negative, [control_2] Positive.  Based
on the overall sentiment expressed in this review, respond with the
relevant control token:" },
  { "role":  "user", "content":  "it's a charming and often
affecting journey." },
  { "role":  "assistant", "content":  "[control_2]" }
],
"image_path":  "none",
}
```

Figure 4: Prompt template for classifying SST-2 movie reviews.

```
Example Prompt (AG News)

 {
"messages":  [
  { "role":  "system", "content":  "You are a classification
expert.  Topics:  [control_1] World, [control_2] Sports, [control_3]
Business, [control_4] Science/Technology.  Based on the content of
this article, respond with the relevant control token:" },
  { "role":  "user", "content":  "Fears for T N pension after
talks.  Unions representing workers at Turner Newall say they are
'disappointed' after talks with stricken parent firm Federal
Mogul." },
  { "role":  "assistant", "content":  "[control_3]" }
],
"image_path":  "none",
}
```

Figure 5: Illustration of an inference prompt from AG News (Business category).

## A.2 EXAMPLE PROMPTS.

## A.3 IMPLEMENTATION DETAILS

We fine-tune all models using a consistent LoRA setup with rank $r = 8$, scaling factor $\alpha = 16$, and dropout of $0.05$. Adapters are applied to the attention projections (q_proj, k_proj, v_proj, o_proj) and to the feed-forward layers (gate_proj, up_proj, down_proj). The token embedding matrix and LM head remain trainable. Under this configuration, the Mistral-3-24B and Gemma-3-27B models expose approximately 1.4B and 1.5B trainable parameters, respectively.

For LM-BFF, we use RoBERTa-base as the backbone, follow the standard prompt-based fine-tuning setup, and use 1 in-context demonstration with 16 training samples per class (consistent with the Gao et al. (2020)). Template and verbalizer settings exactly match those reported in LM-BFF. As expected, LM-BFF performance is sensitive to the number of sampled demonstrations: using more samples improves accuracy but also increases latency substantially.

## A.4 ALTERNATIVE PROMPTING BASELINES

To examine whether simple prompting strategies can emulate the single-symbol classification behavior of our method, we evaluate alternative formulations of the form $[x, \text{cls}, \text{cls-description}]$ using GPT-4o on the MIntRec2.0 benchmark. In this setting, the model is provided with the utterance, associated video frames, and the full mapping between the 30 intent classes and their corresponding

```
Example Prompt (Amazon Reviews)

 {
"messages": [
  { "role":  "system", "content":  "You are a classification
expert.  Topics:  [control_1] Negative, [control_2] Positive.  Based
on the overall sentiment expressed in this review, respond with the
relevant control token:" },
  { "role":  "user", "content":  "DVD Player crapped out after one
year.  I also began having the incorrect disc problems that I've
read about on here.  The VCR still works, but the DVD side is
useless.  I understand that DVD players sometimes just quit on you,
but after not even one year?  To me that's a sign of bad quality.
I'm giving up JVC after this as well.  I'm sticking to Sony or
giving another brand a shot." },
  { "role":  "assistant", "content":  "[control_1]" }
],
"image_path":  "none",
}
```

Figure 6: Illustration of an inference prompt from Amazon Reviews (Negative sentiment).

```
Example Prompt (DBpedia)

 {
"messages": [
  { "role":  "system", "content":  "You are a classification
expert.  Topics:  [control_1] Company, [control_2]
EducationalInstitution, [control_3] Artist, [control_4] Athlete,
[control_5] OfficeHolder, [control_6] MeanOfTransportation,
[control_7] Building, [control_8] NaturalPlace, [control_9] Village,
[control_10] Animal, [control_11] Plant, [control_12] Album,
[control_13] Film, [control_14] WrittenWork.  Based on the content
of this article, respond with the relevant control token:" },
  { "role":  "user", "content":  "Pizza Port Brewing Company is a
brewpub with five locations in Southern California:  Solana Beach,
two in Carlsbad (Downtown and Bressi Ranch), Ocean Beach and San
Clemente.  A former Pizza Port location in San Marcos spun out of
Pizza Port in 2006 and is now an independent operation, the Port
Brewing Company / Lost Abbey brewery.  It has received multiple
awards, including ÿSmall Brewpub of the Yearför both 2003 and 2004
by the Great American Beer Festival and six awards for its beers at
the World Beer Cup." },
  { "role":  "assistant", "content":  "[control_1]" }
],
"image_path":  "none",
}
```

Figure 7: Illustration of an inference prompt from DBpedia (Company category).

indices. The prompt explicitly instructs the model to output only the integer associated with the predicted intent. An example prompt is shown in Figure 9.

Despite the explicit numeric constraint, the model does not consistently produce a single-symbol output. In a non-trivial number of cases (16.67%), GPT-4o generates additional text or multi-token strings. For example, when the expected prediction is 4, the model sometimes outputs a full sentence such as "The intent of the utterance...", or formats the answer as "3: Apologise" rather than emitting a single integer. Furthermore, integers above nine are represented as multi-token sequences under the GPT-4o tokenizer, which inherently increases decoding latency and introduces substantial variability

```
Example of a Classification Training Instance (Topic
Classification)

  {
"messages": [
  { "role": "system", "content": "You are a topic classification
expert. Before making a decision, carefully follow all the
topic-specific instructions/descriptions. Topics: [control_1]
Acknowledge, [control_2] Advise, [control_3] Agree, [control_4]
Apologise, [control_5] Arrange, [control_6] Ask for help, [control_7]
Asking for opinions, [control_8] Care, [control_9] Comfort,
[control_10] Complain, [control_11] Confirm, [control_12] Criticize,
[control_13] Doubt, [control_14] Emphasize, [control_15] Explain,
[control_16] Flaunt, [control_17] Greet, [control_18] Inform,
[control_19] Introduce, [control_20] Invite, [control_21] Joke,
[control_22] Leave, [control_23] Oppose, [control_24] Plan,
[control_25] Praise, [control_26] Prevent, [control_27] Refuse,
[control_28] Taunt, [control_29] Thank, [control_30] Warn. Based on
the above conversation, respond with the relevant topic ID:" },
  { "role": "user", "content": "Thank you so much for your help!
I really appreciate it." },
  { "role": "assistant", "content": "[control_29]" }
],
"image_path": "none",
}
```

Figure 8: Illustration of a prompt from the Topic Classification dataset (Thank intent).

```
Example of a Class-Index Prompting Instance (MIntRec2.0)

  {
"messages": [
  { "role": "system", "content": "You are an intent
classification system. Respond with the number (0--29)
corresponding to the intent. Mapping: 0: Acknowledge, 1:
Advise, ..., 28: Thank, 29: Warn. Output the number (0--29):" },
  { "role": "user", "content": "Thank you" },
  { "role": "assistant", "content": "28" }
],
"image_path": "data:image/jpeg;base64,...",
}
```

Figure 9: Illustration of the class-index prompting baseline evaluated on MIntRec2.0.

in response length. These behaviors prevent the model from operating as a deterministic one-token classifier and stand in contrast to the stable single-token decoding enabled by our learned control-token interface. This analysis indicates that prompting-based alternatives are insufficient to achieve low-latency, single-token inference.

## A.5 LATENCY OF ENCODER-BASED MULTIMODAL BASELINES

Encoder-based multimodal models such as MAG-BERT and MulT use relatively lightweight but rely on a separate vision backbone for per-frame feature extraction. Both pipelines depend on a Swin Transformer ($\sim$88M parameters) for video features.

To quantify this effect, we measure the latency of (i) the Swin Transformer stage and (ii) the MAG-BERT and MulT encoder stages. Table 3 reports median (P50) and tail (P95) latency:

Table 3: Latency breakdown of encoder-based multimodal baselines. The Swin Transformer used for video feature extraction dominates latency, while the encoders themselves are inexpensive.

| Model / Stage | P50 (s) | P95 (s) |
|---|---|---|
| Swin Transformer | 0.23 | 0.40 |
| MAG-BERT | 0.003 | 0.003 |
| MulT | 0.007 | 0.010 |

These results indicate that the encoder components of MAG-BERT and MulT are indeed fast, but the multimodal pipeline is bottlenecked by visual feature extraction. Consequently, end-to-end compute cost cannot be inferred from encoder parameter size alone.

## A.6 ZERO-SHOT ADAPTATION VIA CONTROL-TOKEN PERMUTATIONS

This section examines the model's ability to generalize to unseen mappings between semantic labels and control tokens. During training, the correspondence between labels and control tokens is randomized across instances, preventing the model from relying on fixed token identities. To assess whether the model can operate under entirely new mappings at inference time, we evaluate the fine-tuned **Gemma-3-27B (LaaC)** model on MIntRec 2.0 while applying random permutations of the control-token assignments.

In each of ten runs, a set of 30 control tokens is sampled from the 500-token pool and assigned to the 30 intent labels according to a fresh random permutation. The model is then evaluated on the full test set using this new mapping. This procedure ensures that every run tests the model under a previously unseen association between output tokens and semantic categories. Performance remains highly stable across permutations. The mean accuracy is **44.35%** with a standard deviation of **1.29**, indicating that the model consistently interprets the label descriptions provided in the prompt rather than memorizing token identities. Table 4 summarizes the results.

Table 4: Accuracy under random permutations of control-token assignments during inference.

| Run | Seed | Accuracy (%) |
|---|---|---|
| 1 | 42 | 46.48 |
| 2 | 43 | 44.76 |
| 3 | 44 | 42.94 |
| 4 | 45 | 44.91 |
| 5 | 46 | 42.79 |
| 6 | 47 | 44.61 |
| 7 | 48 | 44.32 |
| 8 | 49 | 44.91 |
| 9 | 50 | 45.70 |
| 10 | 51 | 42.11 |

## A.7 EFFECT OF BATCH SIZE ON LATENCY

This section examines whether the latency benefits of LaaC persist under larger batch sizes, a common setting in practical deployment scenarios (e.g., vLLM-based serving). While the main experiments fix the batch size to 1 for controlled comparison, real-world systems frequently operate with significantly larger batches to maximize throughput. To assess robustness under such conditions, we measure per-sample median (P50) and tail (P95) latencies for batch sizes ranging from 2 to 64 on both Amazon Reviews and MIntRec2.0 using Gemma-3-27B.

Across all batch sizes and for both datasets, LaaC consistently achieves substantially lower latency than the base model. Notably, the improvement remains pronounced even in high-throughput regimes (e.g., batch size 64), where inference engines are typically most optimized. These results demonstrate that the single-token decision mechanism of LaaC yields durable efficiency gains that are insensitive to batching.

**Amazon Reviews.** Table 5 reports per-sample latency across batch sizes. LaaC maintains a clear advantage over the base model for both P50 and P95.

Table 5: Amazon Reviews — Per-sample latency (seconds) vs. batch size for Gemma-3-27B.

| Batch | Base P50 | LaaC P50 (improv.) | Base P95 | LaaC P95 (improv.) |
|---|---|---|---|---|
| 2 | 0.1106 | 0.0545 (**0.0561**↓) | 0.1175 | 0.0875 (**0.0300**↓) |
| 4 | 0.0554 | 0.0327 (**0.0227**↓) | 0.0580 | 0.0427 (**0.0153**↓) |
| 8 | 0.0282 | 0.0205 (**0.0077**↓) | 0.0295 | 0.0222 (**0.0073**↓) |
| 16 | 0.0146 | 0.0108 (**0.0038**↓) | 0.0148 | 0.0110 (**0.0038**↓) |
| 32 | 0.0090 | 0.0068 (**0.0022**↓) | 0.0098 | 0.0069 (**0.0029**↓) |
| 64 | 0.0269 | 0.0056 (**0.0213**↓) | 0.0365 | 0.0053 (**0.0312**↓) |

**MIntRec2.0.** Table 6 shows analogous results in the multimodal setting. The latency reductions are consistent across all batch sizes, indicating that the efficiency advantage of LaaC generalizes beyond text classification to more complex multimodal tasks.

Table 6: MIntRec2.0 — Per-sample latency (seconds) vs. batch size for Gemma-3-27B.

| Batch | Base P50 | LaaC P50 (improv.) | Base P95 | LaaC P95 (improv.) |
|---|---|---|---|---|
| 2 | 0.5197 | 0.4170 (**0.1027**↓) | 0.7818 | 0.6736 (**0.1082**↓) |
| 4 | 0.4343 | 0.3708 (**0.0635**↓) | 0.5966 | 0.5422 (**0.0544**↓) |
| 8 | 0.3917 | 0.3487 (**0.0430**↓) | 0.5179 | 0.4781 (**0.0398**↓) |
| 16 | 0.3670 | 0.3331 (**0.0339**↓) | 0.4424 | 0.4125 (**0.0299**↓) |
| 32 | 0.3488 | 0.3220 (**0.0268**↓) | 0.4069 | 0.3878 (**0.0191**↓) |
| 64 | 0.3451 | 0.3207 (**0.0244**↓) | 0.4111 | 0.3873 (**0.0238**↓) |

We observe that per-sample latency generally decreases as batch size increases. Additionally, these results confirm that the latency improvements of LaaC are not limited to single-example inference, but extend reliably to the large-batch regimes commonly used for high-throughput deployment.

# B ADDITIONAL TEXT CLASSIFICATION BENCHMARKS

To complement the classical text benchmarks used in the main paper (SST-2, Amazon Reviews, AG News, and DBpedia), we further evaluate LaaC on two modern and widely-used datasets (Muennighoff et al., 2022; Enevoldsen et al., 2025): **TweetTopic** (6 classes) (Antypas et al., 2022) and **Banking77** (77 classes) (Casanueva et al., 2020). These datasets are substantially more challenging. As shown in Table 7, LaaC achieves competitive accuracy while providing **14–18× lower latency** than decoder-based generation or GPT-4o. These results further highlight the practicality of LaaC for latency-sensitive text classification.

Table 7: Evaluation results on TweetTopic and Banking77. LaaC models achieve competitive accuracy while providing an order-of-magnitude latency reduction.

| Dataset | Model | Acc. (%) | P50 (s) | P95 (s) |
|---|---|---|---|---|
| | GPT-4o | 77.50 | 0.43 | 0.85 |
| **TweetTopic** | Mistral-3-24B (Base) | 78.13 | 0.24 | 0.27 |
| | **Mistral-3-24B (FT, LaaC)** | **81.86** | **0.03** | **0.05** |
| | GPT-4o | 74.68 | 0.55 | 1.63 |
| **Banking77** | Mistral-3-24B (Base) | 74.74 | 0.17 | 0.33 |
| | **Mistral-3-24B (FT, LaaC)** | 70.94 | **0.03** | **0.03** |

To strengthen our evaluation beyond English-only settings, we further evaluate LaaC on MTOP, a widely used multilingual intent-classification benchmark covering six languages (English, German, Spanish, French, Hindi, Thai).

Table 8: Multilingual evaluation on the MTOP intent classification benchmark (6 languages).

| Model | en | de | es | fr | hi | th | Avg | P50 (s) | P95 (s) |
|---|---|---|---|---|---|---|---|---|---|
| GPT-4o | 79.03 | 78.26 | 77.02 | 79.22 | 74.39 | 78.28 | 77.03 | 0.48 | 1.51 |
| Mistral-3.2-24B (Base) | 81.08 | 79.29 | 81.65 | 80.18 | 77.45 | 70.81 | 78.74 | 0.20 | 0.27 |
| Mistral-3.2-24B (FT, LaaC) | 70.63 | 69.71 | 74.05 | 71.12 | 67.55 | 59.31 | 68.73 | 0.03 | 0.03 |

Since our fine-tuning corpus is entirely English, LaaC naturally achieves lower multilingual accuracy than GPT-4o and the multilingual Mistral base model. Nevertheless, LaaC retains its key advantage—single-token decoding yields over 6× lower latency—highlighting that our classification formulation scales efficiently even in multilingual settings. We expect multilingual accuracy to improve substantially with multilingual supervision, which we identify as an important direction for future work.

## C  PARETO-STYLE ANALYSIS OF ACCURACY, LATENCY, AND GENERALITY

To provide a unified view of model behavior across our three primary objectives—accuracy, latency, and generality—we include a Pareto-style comparison in Figure 10. This visualization situates all evaluated models within a shared three-dimensional space, highlighting the trade-offs among these axes. As shown in the figure, **Gemma-3-27B (LaaC)** occupies the Pareto frontier. It achieves substantially lower latency than GPT-4o while simultaneously attaining higher classification accuracy. Compared to task-specific encoder-based systems such as MAG-BERT, LaaC also achieves markedly higher generality, as measured by cross-domain performance on text benchmarks. This consolidated view illustrates that LaaC delivers improvements across all three dimensions rather than optimizing a single metric in isolation.

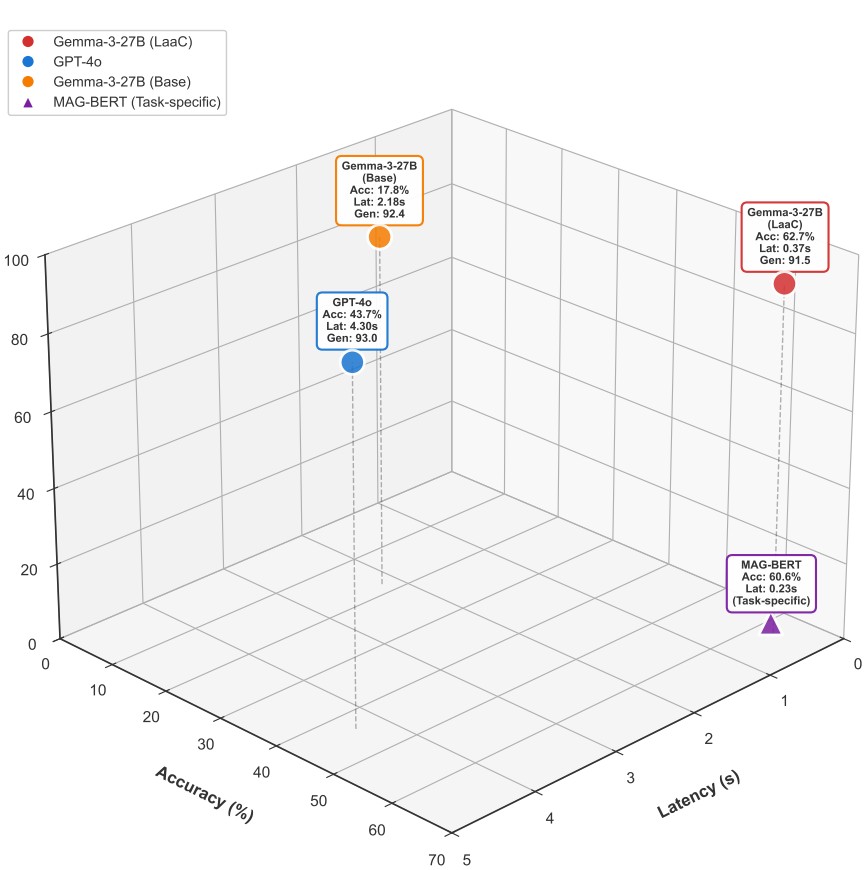

Figure 10: Pareto-style comparison of accuracy, latency, and generality. Models positioned closer to the top-left region exhibit better accuracy–latency trade-offs, while the vertical axis represents generality. Gemma-3-27B (LaaC) lies on the Pareto frontier.

