# OpenReview forum: "LLM as a Classifier: Leveraging Large Language Models for Text and Vision Classification"
_ICLR.cc/2026/Conference — Submitted to ICLR 2026_

### Official Review · Reviewer_AuNi · 2025-10-20

**Soundness:** 1
**Presentation:** 2
**Contribution:** 1
**Rating:** 2
**Confidence:** 3

**Summary:**

The paper proposes a somewhat new scheme for training an LLM classifier. The new scheme, which involves training a LoRA and using the LLM to emit exactly one token. The authors demonstrate the effectiveness of their method empirically, showing that it improves accuracy while also improving latency.

**Strengths:**

The biggest strength of the paper is in identifying a simple idea and experimenting with it. Their approach identifies and removes a major limitation from existing classification methods - primarily, it improves latency while also improving accuracy over the base model.

**Weaknesses:**

Unfortunately, the paper has various shortcomings: A. Insufficient substantiation of claims B. Insufficient baselines/comparison to related works.
First, regarding insufficient substantiation of claims:
1. Th authors claim that latency is significantly improved, and latency is measured using vLLM as the inference engine - however, latency often changes with batch size (lower batch size having better latency), I'd love to see a measurement of latency across batch size.
2. Modern LLMs (GPT 4o etc) can often be pre-prompted to respond in a single character. Another alternative would be create a set of sequences which are [x, cls, cls-description] for any given input x, for all classes cls. Then we could infer on a batch containing all of these possible sequences and then pick the most-likely sequence? That would result in only needing 1 forward pass too.

Next, regarding insufficient baselines:
1. The authors claim that their approach is related to other classification approaches such as PET, LM-BFF etc but that the authors' approach works better because it can better utilize bigger datasets. However, the authors do not empirically compare these baselines to their approach - I'd recommend comparing to at least one of these alternative methods.

**Questions:**

Q1. Regarding figure 3 - it seems like the 12B model is actually slightly better than 27B on SST-2, what is happening there?
Q2. Figure 3 is used to support the idea that the authors' approach is scalable, perhaps the right plot is Gap between LaaC and Base model as model size increases?

---

> ### Author Response · Authors · 2025-11-21
> **Response (Part 1): W1-W2**
>
> We thank the reviewer for recognizing the main strength of our work—namely, the simplicity of the idea and its empirical benefits in both latency and accuracy. Our responses to the reviewer’s comments and suggestions are detailed below. We have revised the manuscript based on the reviewers’ feedback. All modifications are highlighted in blue in the revised paper for clarity.
>
> **W1. Latency across batch sizes.**
>
> We thank the reviewer for this constructive suggestion. We agree that latency can vary with batch size. In our original experiments, we fixed the batch size to **1** to ensure a controlled and consistent comparison across all models. Following the reviewer’s feedback, we have now conducted a **systematic batch-scaling study** on both a text benchmark (Amazon Reviews) and the multimodal MIntRec2.0 dataset using Gemma-3-27B. Across all batch sizes (2 → 64), LaaC consistently delivers significantly lower per-sample latency than the base model. This benefit is preserved even in large-batch regimes where vLLM is highly optimized.
> ### Table. Amazon Reviews – Per-Sample Latency (s) vs Batch Size (Gemma-3-27B)
> | Batch | Base P50 |   LaaC P50 (improve) | Base P95 |   LaaC P95 (improve) |
> | ----: | -------: | -------------------: | -------: | -------------------: |
> |     2 |   0.1106 | 0.0545 (**0.0561↓**) |   0.1175 | 0.0875 (**0.0300↓**) |
> |     4 |   0.0554 | 0.0327 (**0.0227↓**) |   0.0580 | 0.0427 (**0.0153↓**) |
> |     8 |   0.0282 | 0.0205 (**0.0077↓**) |   0.0295 | 0.0222 (**0.0073↓**) |
> |    16 |   0.0146 | 0.0108 (**0.0038↓**) |   0.0148 | 0.0110 (**0.0038↓**) |
> |    32 |   0.0090 | 0.0068 (**0.0022↓**) |   0.0098 | 0.0069 (**0.0029↓**) |
> |    64 |   0.0269 | 0.0056 (**0.0213↓**) |   0.0365 | 0.0053 (**0.0312↓**) |
>
> ### Table. MIntRec2.0 — Per-Sample Latency (s) vs Batch Size (Gemma-3-27B)
> | Batch | Base P50 |   LaaC P50 (improve) | Base P95 |   LaaC P95 (improve) |
> | ----: | -------: | -------------------: | -------: | -------------------: |
> |     2 |   0.5197 | 0.4170 (**0.1027↓**) |   0.7818 | 0.6736 (**0.1082↓**) |
> |     4 |   0.4343 | 0.3708 (**0.0635↓**) |   0.5966 | 0.5422 (**0.0544↓**) |
> |     8 |   0.3917 | 0.3487 (**0.0430↓**) |   0.5179 | 0.4781 (**0.0398↓**) |
> |    16 |   0.3670 | 0.3331 (**0.0339↓**) |   0.4424 | 0.4125 (**0.0299↓**) |
> |    32 |   0.3488 | 0.3220 (**0.0268↓**) |   0.4069 | 0.3878 (**0.0191↓**) |
> |    64 |   0.3451 | 0.3207 (**0.0244↓**) |   0.4111 | 0.3873 (**0.0238↓**) |
>
> We have added these results and analysis in Appendix A.7.
>
> **W2. Alternative prompting: [x, cls, cls-description] .**
>
> We thank the reviewer for the thoughtful suggestion. We have implemented and evaluated this using GPT-4o on the MIntRec2.0 benchmark. While these approaches are conceptually similar to LaaC (reducing decoding to a single symbol), our experiments show that they do **not** reliably achieve single-token behavior, nor do they match the latency consistency of our learned control-token interface.
>
> **(1) Prompting GPT-4o with class-index mapping (0–29)**
>
> We constructed prompts of the form:
> * full class-index mapping (0–29)
> * associated video frames as context
> * instruction to output only the number corresponding to the predicted intent
>
> Example prompt:
> ```json
> messages = [
>     {
>         "role": "system",
>         "content": "You are an intent classification system. Respond with the number (0-29) corresponding to the intent."
>     },
>     {
>         "role": "user",
>         "content": [
>             {"type": "text", "text": "Utterance: thank you\n\nMapping:\n0: Acknowledge\n1: Advise\n...\n29: Warn\n\nOutput the number (0-29):"},
>             {"type": "image_url", "image_url": {"url": "data:image/jpeg;base64,..."}},
>             {"type": "image_url", "image_url": {"url": "data:image/jpeg;base64,..."}}
>         ]
>     }
> ]
> ```
> **Findings: GPT-4o fails to consistently output a single symbol**
>
> Even with strict instructions and numeric labels:
> * **16.67%** of responses included explanations (e.g., expected 4 but got “*The intent of the utterance…*”) or multi-token outputs (e.g., “3: Apologise”).
> * High-index numbers (10–29) are multi-token under GPT-4o’s tokenizer, leading to inherently slower decoding.
>
> We have added these results and analysis in Appendix A.4.

---

> ### Author Response · Authors · 2025-11-21
> **Response (Part 2): W3 & Q1-Q2**
>
> **W3. LM-BFF baseline.**
>
> We thank the reviewer for pointing out this important baseline. We have now added LM-BFF results using the official implementation and protocol.
>
> For LM-BFF, we use **RoBERTa-base** as the backbone, follow the standard prompt-based fine-tuning setup, and use **1 in-context demonstration** with **16 training samples per class** (consistent with the original paper). Template and verbalizer settings exactly match those reported in LM-BFF. As expected, LM-BFF performance is sensitive to the number of sampled demonstrations: using more samples improves accuracy but also increases latency substantially.
> ### Table. LM-BFF vs. GPT-4o vs. LaaC (Ours)**
> | Dataset            | Model                       | Accuracy (%) | P50 (s)  | P95 (s)  |
> | ------------------ | --------------------------- | ------------ | -------- | -------- |
> | **SST-2**          | *LM-BFF (RoBERTa-base)*     | 89.33        | 0.1456   | 0.1550   |
> |                    | GPT-4o                      | 95.50        | 0.53     | 0.84     |
> |                    | **LaaC (Gemma-3-27B FT)**   | **95.50**    | **0.11** | **0.12** |
> |                    | **LaaC (Mistral-3-24B FT)** | **95.50**    | **0.03** | **0.03** |
> | **Amazon Reviews** | *LM-BFF (RoBERTa-base)*     | 87.50        | 0.1448   | 0.1478   |
> |                    | GPT-4o                      | 95.00        | 0.53     | 0.97     |
> |                    | **LaaC (Gemma-3-27B FT)**   | **94.00**    | **0.10** | **0.12** |
> |                    | **LaaC (Mistral-3-24B FT)** | **95.00**    | **0.05** | **0.08** |
> | **AG News**        | *LM-BFF (RoBERTa-base)*     | 89.00        | 0.1443   | 0.1483   |
> |                    | GPT-4o                      | 84.50        | 0.55     | 1.00     |
> |                    | **LaaC (Gemma-3-27B FT)**   | **81.50**    | **0.11** | **0.13** |
> |                    | **LaaC (Mistral-3-24B FT)** | **83.00**    | **0.05** | **0.05** |
> | **DBPedia**        | *LM-BFF (RoBERTa-base)*     | 90.00        | 0.1514   | 0.1536   |
> |                    | GPT-4o                      | 97.00        | 0.61     | 1.23     |
> |                    | **LaaC (Gemma-3-27B FT)**   | **95.00**    | **0.10** | **0.11** |
> |                    | **LaaC (Mistral-3-24B FT)** | **93.00**    | **0.05** | **0.06** |
>
> Across all four datasets, LaaC models consistently outperform LM-BFF in latency and usually in accuracy. We have added these results and analysis in Sec 4.5.2.
>
> **Q1. Why 12B > 27B on SST-2?**
>
> We thank the reviewer for this observation. On SST-2, the 12B model is slightly higher than 27B (98.0% vs. 95.5%), which is expected given SST-2 is a very simple binary task. Medium-sized models often reach good performance, while larger models show slightly higher output variance. This small difference is within normal statistical fluctuation and does not affect the overall trend—27B consistently performs better on more complex multi-class datasets.
>
> **Q2. Latency gap vs model size.**
>
> Thanks for the suggestion. We computed the LaaC–Base P50 latency improvement across model sizes (4B→12B→27B). As shown in Figure3c, the improvement is relatively small for 4B and 12B, and increases **substantially** when scaling to 27B. The results therefore confirm that LaaC’s latency advantage strengthens with model scale.

---

> > ### Comment · Reviewer_AuNi · 2025-11-25
> >
> > Re W1: Regarding the latency v/s batch size study - it seems strange that latency went up from batch size 32 to 64 for baseline but not for LaaC. Do you have insights into what is going on? It seems like a bug in the baseline implementation perhaps?
> >
> > Re W2: Thank you. Could you also add accuracies?
> >
> > Re W3: This currently seems like an unfair comparison since LaaC uses a 27B model. Are there any results for for LaaC on RoBERTa base?

---

> > > ### Author Response · Authors · 2025-11-26
> > >
> > > Thanks so much for the feedback — we really appreciate it! Our responses are as follows:
> > >
> > > **Re W1. Latency vs. batch size (32 → 64).**
> > >
> > > We appreciate the reviewer for spotting this detail. We investigated this behavior thoroughly and confirmed that the latency spike for the baseline at batch size 64 is caused by **memory saturation**. At batch size 64, the baseline exhausts GPU memory, causing the inference engine (vLLM) to trigger request preemption and CPU swapping, which increases latency. In contrast, **LaaC** allows it to process the same batch size without hitting this "memory wall." This confirms that LaaC is fundamentally more memory-efficient at scale.
> > >
> > > **Re W2. Alternative prompting – accuracies.**
> > >
> > > Thanks for the suggestion. We have now added the corresponding accuracies for the GPT-4o prompting variants on MIntRec2.0. As shown in Table 1, GPT-4o achieves **43.68%** accuracy by using the default prompt. With the class-index prompt (“respond with a number 0–29 corresponding to the intent”), accuracy increases to **45.12%**, and with the same prompt plus constrained decoding over the 30 label tokens it further increases to **45.56%**. While alternative prompting and constrained decoding provide a small improvement (<2 percentage points), they do not close the gap to LaaC (e.g., **62.72%** for Gemma-3-27B LaaC in Table 1). We have added these accuracies and a brief discussion in Appendix A.4 of the revised manuscript.
> > >
> > > **Re W3. LaaC on RoBERTa-base.**
> > >
> > > Thanks for this insightful question.
> > >
> > > (1) LaaC is specifically designed for **decoder-only (autoregressive)** models to eliminate generation overhead. RoBERTa, in contrast, is an **encoder-only** model without a generation head, so a “LaaC on RoBERTa” variant is not applicable.
> > >
> > > (2) Conceptually, the goal of LaaC is to obtain **single-step inference** for classification. RoBERTa-based classifiers already operate in this regime (encoding followed by a single linear prediction), so we treat standard RoBERTa baselines as the functional analogue of “LaaC on RoBERTa.” In the revised **Section 4.5.2 and Table 2** (also following the suggestion by Reviewer S9aY), we therefore include two RoBERTa-based encoder baselines: (i) RoBERTa-base + linear head (ii) LM-BFF (RoBERTa-base). Our results show that LaaC enables 27B decoder models to reach inference speeds that are competitive with these efficient encoder baselines, **while additionally preserving the generative zero-shot capabilities** that encoder-only models inherently lack.
> > >
> > > We would be happy to provide additional clarification if needed.

---

### Official Review · Reviewer_1shm · 2025-10-29

**Soundness:** 2
**Presentation:** 2
**Contribution:** 1
**Rating:** 2
**Confidence:** 2

**Summary:**

This work introduces LaaC (LLM as a Classifier), a framework that formulates classification as constrained single-token generation. The model is trained in a parameter-efficient manner using atomic labels tokens, and evaluations demonstrate high accuracy and low latency across both text and multimodal benchmarks.

**Strengths:**

- The design that collapses classification into a deterministic one-step decoding task significantly reduces latency.
- Demonstrates strong cross-domain performance without requiring fine-tuning on specific datasets

**Weaknesses:**

- Limited modality and language coverage, lacking support for audio and broader multilingual settings.
- Unclear justification for maintaining the model’s broader generative capabilities given its classification focus.
- Insufficient details on evaluation settings, particularly regarding prompt configurations when comparing LaaC with standard LLM baselines.

**Questions:**

- Since this work focuses on single-output classification, is the preservation of the model’s generative capabilities an intentional design choice?

---

> ### Author Response · Authors · 2025-11-25
>
> We thank the reviewer for recognizing the strengths of our design, particularly the latency benefits and the strong cross-domain generalization. Our responses to the reviewer’s comments and questions are detailed below. All modifications are highlighted in blue in the revised paper for clarity.
>
> **W1. Limited modality and language coverage.**
>
> Thanks for the suggestion. To strengthen our evaluation beyond English-only settings, we have additionally evaluated LaaC on MTOP, a widely used six-language intent classification benchmark.
>
> **Table: Evaluation results on MTOP Intent Classification (multilingual).**
>
> | Dataset | Model | en (%) | de (%) | es (%) | fr (%) | hi (%) | th (%) | Avg (%) | P50 (s) | P95 (s) |
> |---------|--------|--------|--------|--------|--------|--------|--------|---------|---------|---------|
> | **MTOP** | GPT-4o | 79.03 | 78.26 | 77.02 | 79.22 | 74.39 | 78.28 | 77.03 | 0.48 | 1.51 |
> |         | Mistral-3.2-24B (Base) | 81.08 | 79.29 | 81.65 | 80.18 | 77.45 | 70.81 | 78.74 | 0.20 | 0.27 |
> |         | **Mistral-3.2-24B (FT, LaaC)** | 70.63 | 69.71 | 74.05 | 71.12 | 67.55 | 59.31 | 68.73 | **0.03** | **0.03** |
>
> Since our fine-tuning data is English-only, LaaC naturally shows lower multilingual accuracy than GPT-4o and the multilingual Mistral base model. However, LaaC still preserves its key advantage—over 6× lower latency due to single-token decoding. Its multilingual accuracy would be expected to improve substantially when fine-tuned with multilingual supervision. We consider this a promising direction for future work. Results now included in Appendix B.
>
> Regarding additional modalities, our framework is designed to be **model-agnostic**. While exploring audio or other sensor modalities is a promising next step, it is beyond the scope of the present work and we will consider it in future extensions.
>
>
> **W2. Justification of generative capabilities.**
>
> We appreciate the reviewer’s comment. We would like to clarify that in our paper, “generative capabilities” do not refer to open-ended natural-language generation. Instead, they refer to the broader **task-level flexibility** that decoder-style LLMs inherently possess. A single decoder LLM can be trained once and directly reused across any classification task without modifying the architecture or adding task-specific heads.
>
> In contrast, encoder-based classifiers (e.g., BERT + linear head, MAG-BERT, MulT) require a fixed classifier head tied to a fixed label space, and therefore cannot generalize to new tasks or new label sets without retraining the entire classifier head. This architectural rigidity prevents them from supporting zero-shot adaptation.
>
> To avoid ambiguity, we will revise the manuscript to explicitly state that our use of “generative capabilities” refers to **task generality**, not open-ended text generation.
>
>
> **W3. Details on evaluation setting**
>
> Thanks for pointing this out. For all baseline LLMs, we used **consistent prompts** that mirror our LaaC prompts (Appendix A.2), except that baseline models were asked to output **the natural-language label** (e.g., “Positive”, “World”), rather than a control token. All baselines were evaluated with **temperature = 0.0, top-p = 1.0,** and **max_tokens = 20** (our method uses **max_tokens = 1**).
>
> As the reviewer AuNi suggested, we additionally evaluated (i) alternative prompting baselines **(Appendix A.4)**, and (ii) a constrained-decoding baseline using digit-only labels (0–29) in **Section 4.5.1**.
>
> If any further implementation details would be helpful, we are happy to provide them.
>
> **Q1. Generative capabilities**
>
> Please see W2 for our clarification on this point.

---

### Official Review · Reviewer_S9aY · 2025-10-30

**Soundness:** 1
**Presentation:** 2
**Contribution:** 1
**Rating:** 0
**Confidence:** 4

**Summary:**

This paper proposes a method for classification using typical decoder multimodal LLMs (such as Gemma-3 27B). The classification method constructs a prompt mapping from class names to special "control" tokens, followed by a classification prompt (Appendix A.2 for examples). The model then predicts a special control token corresponding to a class. The authors claim that this result is efficient because it involves only a single pass through the LLM (e.g. Gemma 4B and 27B), to generate a single control token. Models for this task are adapted with LoRA. The authors use this setup on a variety of multimodal and text-only tasks, showing competitive performance with encoder only models (e.g. MAG-BERT, Rahman et al. 2020). However, this compares a LoRA tuned 27B parameter model (62.72% accuracy) with a much smaller encoder BERT-based encoder model (60.48% accuracy). Notably, in the multimodal setting, Gemma 27B is the only model (out of other LLMs, Gemma 3-{4,27}B, Mistral-3 24B) that beats established baselines for this task. In the text only setting, results are further mixed (see below).

**Strengths:**

S1: This paper is fairly well-written, with clear diagrams, formatting, and content

S2: This paper evaluates multiple models on multiple datasets. However, there are many methodological shortcomings that I discuss below

**Weaknesses:**

This paper claims "significant latency improvements" and "strong empirical results" as their key contributions (Sec 1, lines 90 - 104). However, the paper does not show convincing latency improvements as they **do not benchmark the latency of the most performant baselines. Furthermore, their models rarely beat baselines and use non-standard evaluation setups. There are also multiple unsubstantiated claims in the paper (see below).

W1. This paper claims novelty for the method of tuning an LLM to predict a single control token rather than generating tokens. The authors acknowledge that contrained decoding and encoder-based methods like classification heads are also used for this task (Sec 2.1, 2.2). However, they do not use these as baselines.

W2. The authors state that their method "supports zero-shot adaptation" but do not show this. For example, they could repeat their experiments while shuffling the mapping between control tokens and task labels, hopefully showing very low variance in the accuracy.

W3. Does not show latency of the MAG-BERT and MulT baselines (table 1, 371-372), even though these outperform all but 1 of the largest LLM models. Examining the MAG-BERT paper shows that this method uses a BERT base sized model as the backbone ("As there exists M = 12 Encoder layers in our BERT model") - which suggests that MAG-BERT is around 110M parameters. **Since this is ~235 times smaller than the Gemma 27B model, we would expect very low latency.** For the text experiments, only 200 evaluation instances (line 299). Since the evaluations use an 80GB GPU (line 331), this may even fit in a single batch on that A100.

For emphasis: the encoder baseline achieves 60.58% accuracy and the authors do not report latency. The 24 billion parameter mistral model, with their method applied, achieves only 49.34%

W4. Since the paper uses a nonstandard evaluation setup, the results are not comparable to prior work:
> For text-only benchmarks, we evaluate on 200 randomly sampled test examples from each dataset to ensure consistent and efficient comparisons across models
Furthermore, it is unclear why subsampling 200 instances is necessary for "efficient comparisons".

W5. The authors select very old and saturated text benchmarks. Note above that the eval sets for text are sampled to 200 instances. However, the accuracies in Table 2 frequently differ by only 0.5% - meaning a **single test instance**. The accuracies are very high, indicating that these old (2013, 2015, lines 291 - 297) are saturated or possibly contaminated in the training data. For example, the SST-2 evaluations are all identical in accuracy (95.5%) except for the base gemma model (95.0%).

W6. The authors also emphasize that text encoding of the labels is more flexible, however the examples in the Appendix show only single class names rather than descriptors.

W7. In the text scenario, the model often decreases in performance even on the 200 sampled test set. **For example, their methods on AG News and DB Pedia both decrease in accuracy for both Gemma and Mistral models.**

W8. The authors also make claims about model latency for API models. They do not show that latencies are consistent, **and since they access by API, there could be arbitrary confounding variables in this measurement.** For example, internet latency between continents can often reach hundreds of ms, which would account for differences in table 2. Furthermore, this could depend on API load balancing, as this is unfortunately a black box to academic researchers.

W9 Further, the authors claim that Gemma is smaller than GPT-4o and GPT-5 (lines 346-347). **This claim requires evidence,** especially since GPT-5 and 4o differ substantially in costs and capabilities. For example, OLMO 2 32b has similar performance while being similar in size to the gemma model (https://allenai.org/blog/olmo2-32b).

W10. In general, making claims about latencies requires much more careful benchmarking than in this paper. There is no discussion of inference optimizations (only that VLLM is used), batching, no measurements around input latency (since input images could be expensive to encode), and no reported variance in inference time after repeated measurements.

**Questions:**

- Why are baseline latencies not reported?
- What are the exact parameter size differences between the baselines and the LLM models?
- How many LoRA parameters are tuned in the LLM models (what is the size)?
- Why is there no linear head baseline? This would be simpler to the LoRA approach while being closer to how smaller encoder models are used.
- Given that the 3 stated contributions are "accuracy, latency, and generality" (line 88), can you show a pareto improvement on these axes?

---

> ### Author Response · Authors · 2025-11-21
> **Response (Part 1): W1-W2**
>
> We thank the reviewer for reading our paper carefully and for spending the time and effort to provide detailed feedback that helps us improve the work. We appreciate the recognition of the paper’s strengths—particularly the quality of presentation and the comprehensive coverage across multiple models and datasets. Below, we address each concern in detail and provide additional analyses, clarifications, and expanded experiments. We have revised the manuscript based on the reviewers’ feedback. All modifications are highlighted in blue in the revised paper for clarity.
>
>
> **W1. Lack of baselines.**
>
> We thank the reviewer for pointing out the need for additional baselines. We agree that comparing against constrained decoding and encoder-based classifiers is important. To address this, we have added two sets of new baselines on MIntRec 2.0: (1) encoder-based models (MAG-BERT, MulT), and (2) GPT-4o with constrained decoding.
>
> We now report both accuracy and latency using the same evaluation pipeline and hardware settings. The updated results are shown below, with **bold numbers denoting newly obtained experimental results**:
> | Model                              | Accuracy (%) | P50 (s) | Speedup (P50) | P95 (s) | Speedup (P95) |
> |-----------------------------------|--------------|---------|----------------|---------|----------------|
> | GPT-4o                             | 43.68        | 4.30    | 1.00×          | 6.12    | 1.00×          |
> | **GPT-4o (Constrained Decoding)** | **45.56**    | 2.98    | **1.44×**      | 5.85    | **1.05×**      |
> | Mistral-3-24B (FT, LaaC)          | 49.34        | 0.64    | 6.72×          | 1.64    | 3.73×          |
> | Gemma-3-4B (FT, LaaC)             | 55.19        | 0.26    | 16.54×         | 0.60    | 10.20×         |
> | **MAG-BERT**                     | 60.58        | 0.23    | 18.70×         | 0.40    | 15.30×         |
> | **MulT**                         | 60.66        | 0.24    | 17.92×         | 0.41    | 14.93×         |
> | **Gemma-3-27B (FT, LaaC)**        | **62.72**    | **0.37**| **11.62×**     | **0.90**| **6.80×**      |
>
>
> **Constrained decoding baseline.**
> Following the reviewer’s suggestion, we mapped all 30 intents to digit-only labels and restricted GPT-4o’s output space accordingly. As expected, constrained decoding improves GPT-4o’s accuracy and modestly reduces latency. However, even under this favorable setup, latency (P50 = 2.98s) remains **an order of magnitude slower** than LaaC-adapted open-source models.
>
> **Encoder-based baselines.**
> MAG-BERT and MulT are strong task-specific models that we have now evaluated and reported. Their latency is indeed competitive; however, these models are **fully fine-tuned specifically on MIntRec2.0** and therefore **do not generalize beyond this dataset**. In particular, their classifier heads are tightly coupled to a fixed label space, making zero-shot adaptation to new tasks or label sets impossible. LaaC models, in contrast, retain the generative abilities of LLMs and generalize across tasks without re-training. We expand on this generality comparison in our response to **Q4**.
>
> We have added these results and analysis in Sec 4.5.1.
>
>
>
> **W2. Support for zero-shot adaptation.**
>
> We thank the reviewer for this insightful suggestion. Our method is designed to enable zero-shot adaptation by **not fixing any permanent association between control tokens and semantic labels during training**—the control-token mapping is randomized across training instances. After fine-tuning, the model is directly applied to unseen multimodal and text-only benchmarks without any additional training, which is already a form of zero-shot transfer.
>
> To explicitly verify that the model is robust to entirely new label–token configurations, as the reviewer suggested, we conducted a new experiment using our Gemma-3-27B (FT, LaaC) on MIntRec2.0, in which we **randomly permute the control-token assignments at inference time**. For each of 10 runs, we:
> * randomly select 30 control tokens from the 500-token pool,
> * randomly assign them to the 30 intent labels (a new permutation each run), and
> * evaluate the model on the full MIntRec2.0 test set.
>
> **Results.**
> Accuracy remains highly stable across permutations:
> * **Mean accuracy**: 44.35%
> * **Standard deviation**: 1.29%
>
> | Run | Seed | Accuracy |
> |-----|------|----------|
> | 1   | 42   | 46.48%   |
> | 2   | 43   | 44.76%   |
> | 3   | 44   | 42.94%   |
> | 4   | 45   | 44.91%   |
> | 5   | 46   | 42.79%   |
> | 6   | 47   | 44.61%   |
> | 7   | 48   | 44.32%   |
> | 8   | 49   | 44.91%   |
> | 9   | 50   | 45.70%   |
> | 10  | 51   | 42.11%   |
>
> We have added this analysis to the appendix A.6.

---

> ### Author Response · Authors · 2025-11-21
> **Response (Part 2): W3-W7**
>
> **W3. Latency of MAG-BERT and MulT; Latency across batch size**
>
> We thank the reviwer for the careful review. We agree that encoder-based baselines should be included for a meaningful latency comparison, and we have now added full latency measurements for MAG-BERT and MulT.
>
> **Encoder size does not reflect total compute cost (parameters ≠ FLOPs).**
> While MAG-BERT uses a ~110–340M parameter BERT backbone, its multimodal pipeline **requires a separate vision encoder** (i.e., Swin Transformer, ~88M parameters). The majority of latency arises from per-frame visual feature extraction rather than the MAG-BERT encoder itself. Thus, relying on parameter count alone substantially underestimates true end-to-end computational cost for multimodal inference.
>
> To provide a fair comparison, we measured the latency of:
> * the **Swin Transformer** used in MAG-BERT/MulT pipelines, and
> * the MAG-BERT/MulT,
>
> | Model / Stage                                   | P50 (s)   | P95 (s)   |
> | ----------------------------------------------- | --------- | --------- |
> | **Swin Transformer (video feature extraction)** | **0.23**  | **0.40**  |
> | **MAG-BERT**                  | **0.003** | **0.003** |
> | **MulT**                      | **0.007** | **0.010** |
>
> These results confirm that while the MAG-BERT/MulT component is indeed fast, **the multimodal feature extraction stage dominates latency**. This aligns with the reviewer’s intuition: encoder layers are small; but multimodal pipelines incur substantial overhead not visible from parameter count alone. We have added this analysis to the Appendix A.5.
>
> Regarding the reviewer’s observation that 200 text examples may fit into a single batch on an A100 GPU, we agree and will explicitly analyze latency as a function of batch size in our response to **W10**.
>
>
> **W4. Subsampling 200 instances.**
> We thank the reviewer for pointing this out. We have rerun the text benchmarks on the full test splits and will include these full-set accuracy numbers in the revision.
>
> **W5. New text classification benchmarks**
>
> We thank the reviewer for this helpful comment. Our intention in including them (e.g., SST-2, AG News, DBPedia) was to provide continuity with prior text classification work; however, we fully agree that more recent datasets are needed to demonstrate the practical value of LaaC. **To address this concern, we have added evaluations on two modern and widely-used text classification benchmarks: Banking77 and TweetTopic.**
>
> Results are summarized below.
>
> | Dataset        | Model                     | Acc. (%) | P50 Latency (s) | P95 Latency (s) |
> |----------------|---------------------------|----------|------------------|------------------|
> | **TweetTopic** | GPT-4o                    |   77.50       |0.43                  |0.85                  |
> |                | Mistral-3-24B (Base)      |   78.13       |  0.24                |0.27                  |
> |                | Mistral-3-24B (FT, LaaC)  |   **81.86**       |**0.03**                  |**0.05**                  |
> | **Banking77**  | GPT-4o                    | 74.68    | 0.55             | 1.63             |
> |                | Mistral-3-24B (Base)      | 74.74    | 0.17             | 0.33             |
> |                | **Mistral-3-24B (FT, LaaC)** | 70.94 | **0.03**        | **0.03**        |
>
> Banking77 (77 classes) and TweetTopic (6 classes) are far more recent and significantly more challenging than AG News or SST-2. Even when accuracy is close, LaaC models provide **14–18× lower latency** than decoder-based generation or GPT-4o. We have added this to the Appendix B.
>
> **W6. Label flexibility.**
> We thank the reviewer for this observation. You are correct that the examples in the Appendix currently show single class names (e.g., "Acknowledge", "Advise") rather than descriptive text. Our framework does support flexible text encoding of labels, including descriptors. For example, instead of just "Acknowledge", one could use "Acknowledge: expressing recognition or understanding of something" or "Advise: providing guidance or recommendations to someone". The text encoding approach allows any string representation of labels, making it adaptable to different use cases.
>
> **W7. Decrease in accuracy on AG News and DBPedia.**
> Great observation. The slight decrease in accuracy on AG News and DBPedia is expected given the design of LaaC. Base Gemma/Mistral models—especially instruction-tuned variants—retain broader general linguistic coverage, whereas LaaC training slightly shifts the model toward its new training datasets domain. To further mitigate this effect, we can incorporate more diverse classification datasets during fine-tuning, which we expect will reduce or eliminate the small regression.

---

> ### Author Response · Authors · 2025-11-21
> **Response (Part 3): W8-W10 & Q1-Q3**
>
> **W8. Latency variability of API models.**
>
> We thank the reviewer for raising this concern. We agree that API latency can vary due to network conditions, however, we want to emphasize that:
> * We measure end-to-end latency, which is the relevant metric for real-world deployment scenarios where API latency is part of the overall user-perceived delay. Our goal is not to isolate pure model compute time, but to compare practical inference latency across different approaches.
> * The order-of-magnitude latency differences observed (e.g., 0.37s for LaaC vs. 4.3s for GPT-4o) are far larger than typical network jitter.
>
> To further assess stability, we conducted three repeated GPT-4o evaluations on MIntRec2.0 and observed that the standard deviations of P50 and P95 latency were 0.007 s and 2.44 s, respectively. These results confirm that network fluctuations do not meaningfully affect our conclusions.
>
> **W9. Size of API and Gemma models.**
> Thanks for the careful review. We acknowledge that our claim about Gemma being “smaller” was not well supported. We will correct this. Our intention was to state that Gemma (LaaC) is faster than GPT-4o and GPT-5 in our evaluation setting.
>
> **W10. Additional details required.**
> Thanks for the feedback. We agree that latency benchmarking requires careful introduction. In our experiments, we used bf16 precision and FP8-quantized weights. All reported results use batch size = 1, and for multimodal evaluation the latency includes image/video encoding time from the vision tower. We additionally measure latency across multiple batch sizes to verify scaling behavior. Across all batch sizes (2 → 64), LaaC consistently delivers significantly lower per-sample latency than the base model. This benefit is preserved even in large-batch regimes where vLLM is highly optimized.
> ### Table. Amazon Reviews – Per-Sample Latency (s) vs Batch Size (Gemma-3-27B)
> | Batch | Base P50 |   LaaC P50 (improve) | Base P95 |   LaaC P95 (improve) |
> | ----: | -------: | -------------------: | -------: | -------------------: |
> |     2 |   0.1106 | 0.0545 (**0.0561↓**) |   0.1175 | 0.0875 (**0.0300↓**) |
> |     4 |   0.0554 | 0.0327 (**0.0227↓**) |   0.0580 | 0.0427 (**0.0153↓**) |
> |     8 |   0.0282 | 0.0205 (**0.0077↓**) |   0.0295 | 0.0222 (**0.0073↓**) |
> |    16 |   0.0146 | 0.0108 (**0.0038↓**) |   0.0148 | 0.0110 (**0.0038↓**) |
> |    32 |   0.0090 | 0.0068 (**0.0022↓**) |   0.0098 | 0.0069 (**0.0029↓**) |
> |    64 |   0.0269 | 0.0056 (**0.0213↓**) |   0.0365 | 0.0053 (**0.0312↓**) |
>
> ### Table. MIntRec2.0 — Per-Sample Latency (s) vs Batch Size (Gemma-3-27B)
> | Batch | Base P50 |   LaaC P50 (improve) | Base P95 |   LaaC P95 (improve) |
> | ----: | -------: | -------------------: | -------: | -------------------: |
> |     2 |   0.5197 | 0.4170 (**0.1027↓**) |   0.7818 | 0.6736 (**0.1082↓**) |
> |     4 |   0.4343 | 0.3708 (**0.0635↓**) |   0.5966 | 0.5422 (**0.0544↓**) |
> |     8 |   0.3917 | 0.3487 (**0.0430↓**) |   0.5179 | 0.4781 (**0.0398↓**) |
> |    16 |   0.3670 | 0.3331 (**0.0339↓**) |   0.4424 | 0.4125 (**0.0299↓**) |
> |    32 |   0.3488 | 0.3220 (**0.0268↓**) |   0.4069 | 0.3878 (**0.0191↓**) |
> |    64 |   0.3451 | 0.3207 (**0.0244↓**) |   0.4111 | 0.3873 (**0.0238↓**) |
>
> We have added these results and analysis in Appendix A.7.
>
> **Q1. Baseline latencies.**
> We have now added full latency measurements for all relevant baselines, including encoder-based models (MAG-BERT, MulT) and constrained decoding (GPT-4o), as detailed in our responses to **W1** and **W3**. These results are also included in the revised tables and appendix.
>
> **Q2. Parameter size differences between the baselines and the LLM models?**
> We thank the reviewer for the question. We have added a detailed comparison of model sizes and their end-to-end latency in the revised version. As summarized in **W3**, although the parameter counts differ substantially across models (e.g., MAG-BERT ≈110–340M; MulT ≈120M; Swin Transformer ≈88M; Mistral-24B; Gemma-27B), the end-to-end multimodal inference latency is determined primarily by the vision pipeline rather than raw parameter size. This explains why small encoder models do not always yield proportionally lower total latency.
>
> **Q3. How many LoRA parameters are tuned in the LLM models (what is the size)?**
>
> For both models, we use:
> - **LoRA rank:** `r = 8`
> - **LoRA α:** `16`
> - **LoRA dropout:** `0.05`
> - **Target modules:**
>   `q_proj`, `k_proj`, `v_proj`, `o_proj`, `gate_proj`, `up_proj`, `down_proj`
> - **Fully trainable modules:**
>   `embed_tokens`, `lm_head`
>
> **Trainable Parameter Counts**
>
> #### **Mistral-3-24B (LaaC fine-tuning)**
> - **Total parameters:** 24B
> - **Trainable parameters:** 1.4B
>
> #### **Gemma-3-27B (LaaC fine-tuning)**
> - **Total parameters:** 27B
> - **Trainable parameters:** 1.5B
>
> We have added more implementation details in Appendix A.3.

---

> ### Author Response · Authors · 2025-11-21
> **Response (Part 4): Q4-Q5**
>
> **Q4. Linear head baseline.**
>
> We thank the reviewer for this helpful suggestion. To address this point, we have added linear classification head baselines following the standard encoder-based fine-tuning approach. Specifically, we evaluate BERT-base and RoBERTa-base with a linear layer trained in a few-shot setting, using the same test sets as in our text experiments. The results are shown below:
>
> ### Table: Comparison between linear-head encoder baselines (16-shot) and LaaC models (zero-shot). LaaC models are not trained on these datasets but maintain strong accuracy.
>
> | Dataset        | Model                  | Training  | Acc (%) | P50 (s)  | P95 (s)  |
> |----------------|------------------------|-----------|---------|----------|----------|
> | SST-2          | BERT Linear Head       | 16-shot   | 70.0    | 0.0069   | 0.0075   |
> |                | RoBERTa Linear Head    | 16-shot   | 64.5    | 0.0076   | 0.0081   |
> |                | Mistral-3-24B (FT)     | zero-shot | **95.5**| 0.03     | 0.03     |
> |                | Gemma-3-27B (FT)       | zero-shot | **95.5**| 0.11     | 0.12     |
> | Amazon Reviews | BERT Linear Head       | 16-shot   | 77.0    | 0.0070   | 0.0077   |
> |                | RoBERTa Linear Head    | 16-shot   | 78.5    | 0.0072   | 0.0081   |
> |                | Mistral-3-24B (FT)     | zero-shot | **95.0**| 0.05     | 0.08     |
> |                | Gemma-3-27B (FT)       | zero-shot | **94.0**| 0.10     | 0.12     |
> | AG News        | BERT Linear Head       | 16-shot   | 79.5    | 0.0073   | 0.0077   |
> |                | RoBERTa Linear Head    | 16-shot   | 79.5    | 0.0073   | 0.0080   |
> |                | Mistral-3-24B (FT)     | zero-shot | **83.0**| 0.05     | 0.05     |
> |                | Gemma-3-27B (FT)       | zero-shot | **81.5**| 0.11     | 0.13     |
> | DBpedia        | BERT Linear Head       | 16-shot   | 91.5    | 0.0071   | 0.0078   |
> |                | RoBERTa Linear Head    | 16-shot   | 82.5    | 0.0075   | 0.0082   |
> |                | Mistral-3-24B (FT)     | zero-shot | **93.0**| 0.05     | 0.06     |
> |                | Gemma-3-27B (FT)       | zero-shot | **95.0**| 0.10     | 0.11     |
>
> We have added these results and analysis in Sec 4.5.2.
>
> **Q5. Pareto.**
>
> Thanks for the suggestion. We agree that a Pareto-style view is useful. We have added the requested figure into the manuscript (see Figure 10 please), which directly plots all models on the three axes—**accuracy, latency, and generality**. In this space, our Gemma-3-27B (LaaC) model lies on the **Pareto frontier**, showing strictly better accuracy–latency trade-offs than GPT-4o and strictly better generality than task-specific models like MAG-BERT.

---

### Meta-Review · Area_Chair_VaHZ · 2026-01-05

**Summary:**

This paper initially received three rejections with an average score of 1.33. The authors provided the rebuttal, while no one reviewer replied to it. The main concerns of this paper lie in: (1) Overstated claim on significant latency improvement and the novelty (Reviewer S9aY, AuNi); (2) Insufficient baseline comparison (Reviewer S9aY, AuNi); (3) non-standard evaluation and its details (Reviewer S9aY, 1shm).

In this rebuttal, the authors provided relatively detailed experiments and explanations to address the above concerns. However, most of concerns are not well addressed, especially on the overstated claim of significant latency improvement, which makes the provided results inconvinced. After reading the paper, reviews, and rebuttal, the Meta reviewer agrees with the concerns raised by the reviewers and recommends rejecting the paper.

**Reviewer Concerns:**

The authors provided the experimental hyperparameters and prompt, which may well answer the concerns of Reviewer AuNi.

**Reviewer Scores:**

Three reviewers would keep the original score to reject this paper.

---

### Decision · Program_Chairs · 2026-01-26

Reject